# Disrupting hierarchical control of nitrogen fixation enables carbon-dependent regulation of ammonia excretion in soil diazotrophs

**Marcelo Bueno Batista**[1]*, **Paul Brett**[2], **Corinne Appia-Ayme**[1], **Yi-Ping Wang**[3]*,
**Ray Dixon**[1]*

**1** Department of Molecular Microbiology, John Innes Centre, Norwich, United Kingdom, **2** Department of Metabolic Biology, John Innes Centre, Norwich, United Kingdom, **3** State Key Laboratory of Protein and Plant Gene Research, School of Life Sciences & School of Advanced Agricultural Sciences, Peking University, Beijing, China

* marcelo.batista@jic.ac.uk (MBB); wangyp@pku.edu.cn (YPW); ray.dixon@jic.ac.uk (RD)

**Data Availability Statement:** All relevant data are within the manuscript and its Supporting Information files.

## Abstract

The energetic requirements for biological nitrogen fixation necessitate stringent regulation of this process in response to diverse environmental constraints. To ensure that the nitrogen fixation machinery is expressed only under appropriate physiological conditions, the dedicated NifL-NifA regulatory system, prevalent in Proteobacteria, plays a crucial role in integrating signals of the oxygen, carbon and nitrogen status to control transcription of nitrogen fixation (*nif*) genes. Greater understanding of the intricate molecular mechanisms driving transcriptional control of *nif* genes may provide a blueprint for engineering diazotrophs that associate with cereals. In this study, we investigated the properties of a single amino acid substitution in NifA, (NifA-E356K) which disrupts the hierarchy of *nif* regulation in response to carbon and nitrogen status in *Azotobacter vinelandii*. The NifA-E356K substitution enabled overexpression of nitrogenase in the presence of excess fixed nitrogen and release of ammonia outside the cell. However, both of these properties were conditional upon the nature of the carbon source. Our studies reveal that the uncoupling of nitrogen fixation from its assimilation is likely to result from feedback regulation of glutamine synthetase, allowing surplus fixed nitrogen to be excreted. Reciprocal substitutions in NifA from other Proteobacteria yielded similar properties to the *A. vinelandii* counterpart, suggesting that this variant protein may facilitate engineering of carbon source-dependent ammonia excretion amongst diverse members of this family.

## Author summary

The NifL-NifA regulatory system provides dedicated signal transduction machinery to regulate nitrogen fixation in diverse Proteobacteria. Understanding how the balance of nitrogen and carbon resources is signalled via NifL-NifA for precise control of nitrogen fixation may lead to broadly applicable translational outputs. Here, we characterize a NifA variant that bypasses nitrogen regulation but is still dependent on the carbon status to

**Funding:** This study was supported by the UKRI Biotechnology and Biological Sciences Research Council, (https://www.ukri.org/) (grants BB/N013476/1 and BB/N003608/1 to R.D), the Royal Society (https://royalsociety.org/) (ICA\R1\180088 to R.D.), the National Science Foundation of China (NSFC), (http://www.nsfc.gov.cn) (grant 32020103002 to Y-P.W), and the National Key R&D Program of China (http://www.most.gov.cn) (grant 2019YFA0904700 to Y-P.W). The funders had no role in study design, data collection and analysis, decision to publish, or preparation of the manuscript.

**Competing interests:** The authors have declared that no competing interests exist

enable ammonia excretion in soil diazotrophs. Disruption of the regulatory hierarchy in response to nitrogen and carbon suggests how the integration of environmental stimuli could be harnessed to engineer conditional release of fixed nitrogen for the benefit of cereal crops.

## Introduction

Biological nitrogen fixation requires diversion of reducing equivalents and ATP derived from carbon metabolism to support the high energetic demands of the enzyme nitrogenase, which converts dinitrogen to ammonia. Tight regulatory control provides the means to balance energy metabolism with nitrogen fixation and ammonia assimilation so that fixed nitrogen is not limiting under diazotrophic growth conditions. Achieving an appropriate balance between carbon and nitrogen metabolism is particularly important for diazotrophic bacteria in order to meet the energetic cost of nitrogen fixation, while thriving in competitive environments. While many studies on the regulation of nitrogen fixation have focused on intricate signalling mechanisms responding to the presence of oxygen and fixed nitrogen [1–3], regulation in response to the carbon status has not been extensively studied, despite its significance for the energetics of diazotrophy and the interplay required to balance the carbon: nitrogen ratio.

The nitrogen fixation specific, NifL-NifA, regulatory system provides a very sophisticated signal transduction complex for integration and transmission of various environmental cues to the transcriptional apparatus in *Azotobacter vinelandii* to regulate biosynthesis and expression of nitrogenase, reviewed in [4,5]. The anti-activator NifL is a multidomain protein carrying an N-terminal PAS domain (PAS1) that senses the redox status via a FAD co-factor [6,7]. A second PAS domain (PAS2) appears to play a structural role in relaying the redox changes perceived by the PAS1 domain to the central (H) and C-terminal (GHKL) domains of NifL [8,9]. The latter is responsible for ADP binding [10,11] and is probably the site of interaction for the GlnK signal transduction protein, allowing integration of the nitrogen input into NifL-NifA regulation [12,13]. The protein partner of NifL, the prokaryotic enhancer binding protein NifA, which activates *nif* transcription, is comprised of an N-terminal regulatory domain (GAF), a central AAA+ sigma-54 activation domain and a C-terminal DNA binding domain. The regulatory GAF domain of *A. vinelandii* NifA binds 2-oxoglutarate [14,15], a TCA cycle intermediate at the interface of carbon and nitrogen metabolism [16]. NifA can only escape inhibition by NifL, when the GAF domain is saturated with 2-oxoglutarate, thus potentially providing a mechanism for the NifL-NifA system to respond to the carbon status.

Understanding how the NifL-NifA system integrates diverse regulatory inputs may allow new strategies for engineering diazotrophs with enhanced ability to fix nitrogen and release ammonia to benefit crop nutrition. Ammonia excretion can be achieved in *A. vinelandii* by engineering constitutive expression of genes required for nitrogenase biosynthesis through inactivation of *nifL* [17–20]. However, in the absence of active NifL, all the regulatory signal inputs that control NifA activity are removed, and as consequence, the resulting bacterial strain may be severely disadvantaged in the environment and even unstable under laboratory conditions as already reported [18,20]. Isolation of insertion mutants in *nifL* appears to be conditional upon second site mutations that may alter the level of *nif* gene expression. One such suppressor was identified in a promoter-like sequence upstream of *nifA*, presumably leading to a permissive reduction in *nifA* transcript levels [20]. A full deletion of *nifL* has also been reported [19], but it is unclear if second site mutations occurred during its isolation.

One approach to potentially minimize the energetic burden associated with constitutive expression of nitrogenase is to ensure that regulatory control of NifA activity is maintained in

energy-limiting environments. Random mutagenesis of *nifA* followed by screening the activity of the *A. vinelandii* NifL-NifA system in *E. coli* identified various NifA variants able to escape regulation by NifL under nitrogen excess conditions [21]. One of these mutations, resulting in a charge-change substitution, E356K, located in the central catalytic domain of NifA, (hereafter named NifA-E356K), was found to require binding of 2-oxoglutarate to the GAF domain to escape NifL repression in response to excess fixed nitrogen [15,22].

In this study we demonstrate that both expression and activity of nitrogenase are insensitive to the nitrogen status when the *nifA-E356K* mutation is introduced into *A. vinelandii*, resulting in excretion of ammonia at millimolar levels during exponential growth, a phenomenon correlated with feedback regulation of glutamine synthetase when nitrogenase is constitutively active. However, unregulated expression of *nif* genes and ammonia excretion by the *nifA-E356K* mutant is conditional on the nature of the carbon source indicating dependency on carbon status signalling and supporting previous biochemical observations that this NifA-E356K variant is dependent on the levels of 2-oxoglutarate to escape nitrogen regulation by NifL *in vitro* [22]. Finally, we demonstrate that reciprocal substitutions in NifA proteins of other Proteobacteria lead to similar regulatory phenotypes when assayed in *E. coli* as a chassis and also when the substitution is engineered in the endophytic diazotroph *Pseudomonas stutzeri* A1501 [23,24]. In principle, this single amino acid substitution in NifA (E356K) provides a regulatory switch capable of activating *nif* gene expression under nitrogen excess conditions only when certain carbon sources are available in the environment. This sets the foundation for engineering a synthetic symbiosis, in which ammonia excretion by cereal associative diazotrophs is conditionally regulated in order to deliver fixed nitrogen to cereal crops.

## Results

### The activity of NifA-E356K is not regulated in response to the nitrogen status in *A. vinelandii*, resulting in ammonia excretion

Previously the NifA-E356K variant protein was characterized either *in vivo* using *E. coli* as a chassis, or *in vitro* using purified protein components [21,22]. To evaluate if this variant protein would bypass NifL regulation in the original *A. vinelandii* DJ background, we introduced the *nifA-E356K* mutation into the chromosome (strain EK) and examined its influence on transcriptional regulation of the nitrogenase structural genes using RT-qPCR. As anticipated, no *nifH* transcripts were detected in the wild type strain (DJ) in the presence of either 25 or 5 mM of ammonium acetate, whilst increased transcript levels were observed in the absence of ammonium (Fig 1A). In contrast, high levels of *nifH* transcripts were observed in all conditions tested for the *nifA-E356K* mutant strain (EK) (Fig 1B), confirming that *nifA-E356K* is able to escape nitrogen regulation in *A. vinelandii*. Comparison of the levels of *nifH*, *nifL* and *nifA* transcripts in the EK strain relative to the wild type (DJ) (Fig 1C) revealed that whilst *nifH* levels are higher in the EK mutant, this is not correlated with increased levels of *nifL* and *nifA* transcripts. This is in line with previous reports establishing that *nifLA* expression is not subject to autoactivation by NifA in *A. vinelandii* [25,26] and suggests that constitutive expression of the nitrogenase structural gene operon in the EK mutant is intrinsic to the *nifA-E356K* mutation itself, rather than a consequence of overexpression of this mutant gene. The increase in *nifH* transcripts correlated with higher nitrogenase activity in the EK strain, which was not regulated in response to excess fixed nitrogen, in contrast to the wild-type strain (Fig 1D). As anticipated from the energetic constraints associated with constitutive expression and activity of nitrogenase, the EK mutant had an apparent growth deficiency in liquid media (S1 Fig). However, when an insertion replacing the *nifH* structural gene was introduced into the EK strain to generate strain EKH (EK,

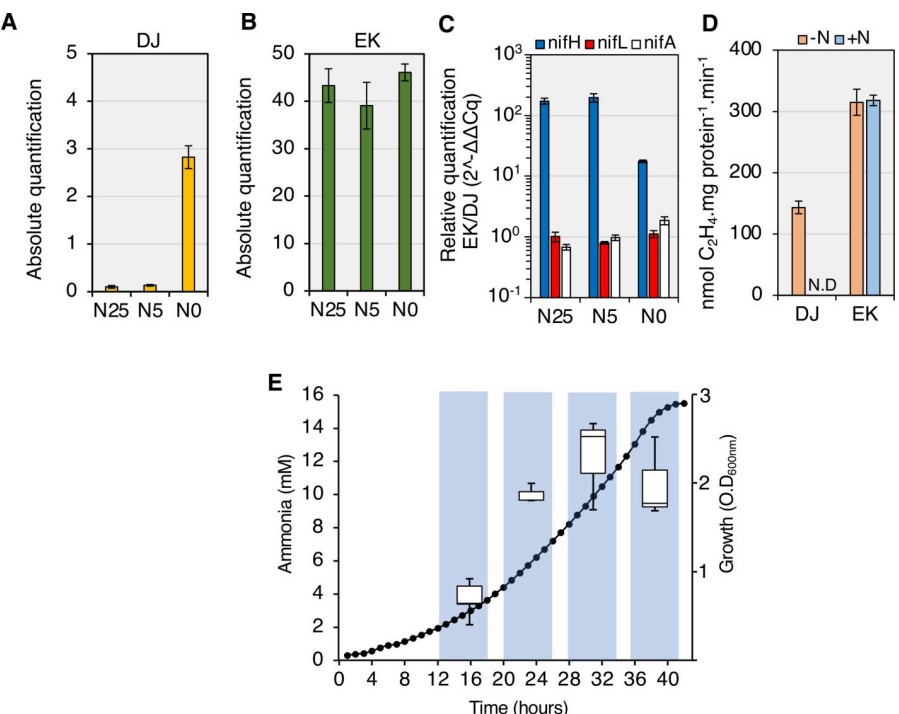

**Fig 1. Constitutive expression and activity of nitrogenase results in ammonia excretion by the *nifA-E356K* strain (EK) with sucrose as the carbon source.** (A) Absolute levels of *nifH* transcripts in the wild type (DJ) and (B) *nifA-E356K* (EK) under three different nitrogen regimes: 25 mM (N25), 5 mM (N5) or 0 mM (N0) ammonium acetate. (C) Relative levels of *nifH*, *nifL* and *nifA* transcripts between the strains EK and DJ. The graph is presented on a $log_{10}$ scale to emphasize that the relative levels of *nifL* and *nifA* transcripts are close to 1 in all conditions. (D) *In vivo* nitrogenase specific activities in the absence (-N) or presence (+N) of 20 mM ammonium chloride. Activity was determined by the acetylene reduction assay using cultures grown to an O.D$_{600nm}$ between 0.3–0.4 as described in the methods. N.D: not detected. (E) Ammonia from the culture supernatant was quantified in the EK strain (left y axis, box blots) in the growth phases indicated by the bars shaded in blue (right y axis, closed circles).

*ΔnifH*::*tetA)* this growth penalty was alleviated (S1D–S1F Fig) suggesting that it results from unregulated expression and activity of nitrogenase.

When cultivated under diazotrophic conditions in nitrogen-free media supplemented with sucrose as carbon source, the EK strain excreted millimolar levels of ammonia (Fig 1E). The onset of ammonia excretion occurred at the early stages of growth (O.D$_{600nm}$ 0.3–0.6) reaching a peak at mid to late-exponential phase (O.D$_{600nm}$ 1.4–2.0) and declined upon entry into stationary phase, potentially as a consequence of oxygen limitation and lower nitrogen fixation rates.

## Nitrogen fixation and ammonium assimilation are uncoupled in the *nifA-E356K* mutant strain

It is perhaps surprising that a single amino acid substitution in NifA enables ammonia excretion, since lowering the flux of ammonia assimilation through the glutamine synthetase-glutamate synthase (GS-GOGAT) pathway is expected to be an additional pre-requisite for high level release of ammonia. The enzyme glutamine synthetase (GS) is a key component for ammonia assimilation in proteobacteria and is subject to post-translational regulation via adenylylation in response to the nitrogen status [27,28]. Under excess nitrogen conditions, GS is adenylylated reducing its biosynthetic activity [3,28]. Comparison of the Mg$^{2+}$, ATP-dependent glutamine

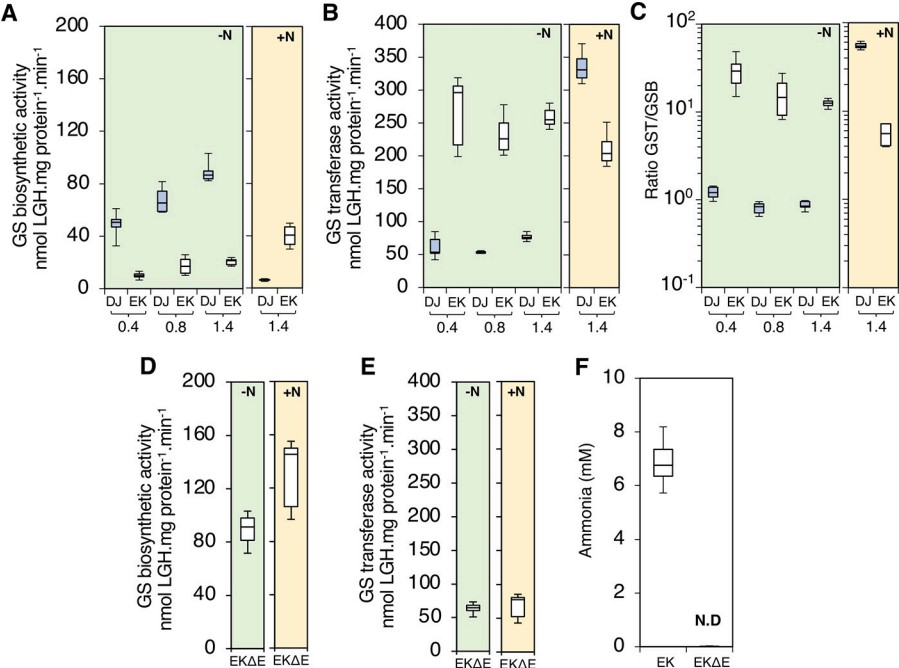

**Fig 2. Ammonia excretion is dependent upon lower glutamine synthetase biosynthetic activity in the *nifA-E356K* strain (EK).** (A) Glutamine synthetase (GS) biosynthetic and (B) transferase activities were measured in the wild type (DJ, blue box plots) and E356K (EK, white box plots) strains in three different phases of growth, corresponding to an $O.D_{600\,nm}$ of 0.4, 0.8 and 1.4 as indicated. Ratio between GS transferase (GST) and GS biosynthetic (GSB) activities is presented in (C) on a $log_{10}$ scale to emphasize that the ratio between GST and GSB activities are close to 1 in the wild type (DJ) in all growth phases. (D) Glutamine synthetase (GS) biosynthetic and (E) transferase activities measured in the EKΔE strain (*nifA-E356K* with a *glnE* deletion) at an $O.D_{600\,nm}$ of 0.8. The charts shaded in green represent the activities under diazotrophic conditions (-N), while those shaded in yellow represent the activity in the presence of excess fixed nitrogen, (20 mM $NH_4Cl$, +N). (F) Ammonia from the culture supernatant was quantified in both the EK and EKΔE strains grown under diazotrophic conditions. N.D: not detected.

synthetase biosynthetic (GSB) and the $Mn^{2+}$, AsO4-, ADP-dependent glutamine transferase (GST) activities can provide a snapshot of GS adenylylation states *in vivo*, given that GSB activity can only be detected in the non-adenylylated enzyme subunits [29–31]. We observed that GSB activity was higher in wild type (DJ) when compared to the mutant (EK) in all growth phases when cells were cultivated under diazotrophic conditions (green shaded plots in Fig 2A). Conversely, GST activity was higher in the *nifA-E356K* strain (EK) than in the wild type (DJ) (Fig 2B). A comparison of GST/GSB ratios in both strains (Fig 2C) demonstrated that GS in the wild type (DJ) is likely to be entirely non-adenylylated under diazotrophic conditions (GST/GSB ratio ≦ 1), whilst the NifA-E356K mutant (EK) had much higher GST/GSB ratios (ranging from 15 to 30-fold) suggesting that in this mutant strain, GS is more heavily adenylylated. Under fixed nitrogen excess conditions (yellow shaded plots in Fig 2), as anticipated, a marked reduction of GSB activity was observed in the DJ strain (Fig 2A, yellow plot) followed by an increase in the GST activity (Fig 2B, yellow plot). For the EK strain however, relatively minor changes in GSB and GST activities occurred in response to the nitrogen status. A slight increase in GSB activity was observed in the EK strain under nitrogen excess conditions, but this was not apparently associated with changes in the expression of GS (*glnA*) itself (S2 Fig). Taken together, these results confirm that the ability of the *nifA-E356K* mutant strain to assimilate ammonia via GS is reduced compared to the wild type under diazotrophic conditions. This is likely to be a consequence of increased

adenylylation by the bifunctional adenylyl transferase enzyme GlnE, which carries out post-translational modification of GS in response to the nitrogen status. Since deletion of *glnE* prevents adenylylation of GS in *A. vinelandii* [32], we introduced a *glnE* deletion into the *nifA-E356K* background to generate the strain EKΔE. Under diazotrophic conditions, GSB and GST activities in the EKΔE strain were similar to the those in the wild type strain (DJ) and clearly distinct from the activities in the *nifA-E356K* strain (EK) (compare green shaded plots in Fig 2A, 2B, 2D and 2E). However, as anticipated from the absence of adenylyl transferase activity, addition of ammonium to the media did not lead to either reduction of GSB or increased GST activity in the EKΔE strain when compared to the DJ strain (compare yellow shaded plots in Fig 2A, 2B, 2D and 2E). As expected from the increased GS biosynthetic activity exhibited by the EKΔE strain under nitrogen excess conditions, ammonia excretion was ablated in this strain (Figs 2F and S3). These results therefore imply that the ability of the *nifA-E356K* (EK) strain to excrete ammonia is dependent upon feedback regulation, resulting in constitutive adenylylation of GS and hence decreased ammonia assimilation.

## Carbon signalling dominates regulation of *nif* gene expression in the *nifA-E356K* strain

All the above experiments were conducted in media containing sucrose, a carbon source enabling a high flux through the TCA cycle that evidently maintains sufficient 2-oxoglutarate to activate NifA in *A. vinelandii*. Although the potential for carbon regulation, signalled via binding of 2-oxoglutarate to the NifA GAF domain, has been well established *in vitro*, the physiological relevance of 2-oxoglutarate in NifL-NifA regulation has not been clearly demonstrated *in vivo*. NifA-E356K requires 2-oxoglutarate in order to escape inhibition by NifL in the presence of GlnK *in vitro* [15,22], suggesting that its ability to bypass nitrogen regulation *in vivo* might be regulated by carbon source availability. To facilitate correlation of nitrogenase activity with expression of the nitrogenase structural genes when strains were grown on different carbon sources we constructed strains containing a translational *nifH*::*lacZ* fusion located at a neutral site in the chromosome (see Materials and Methods and S1 Table). Initial screening for the ability of the NifA-E356K protein to escape nitrogen regulation in several carbon sources (S4 Fig) revealed that this variant protein supported strong activation of the *nifH* promoter under nitrogen excess conditions when grown in sucrose, glucose or glycerol. In contrast, only approximately 20–40% of the maximum activity was observed when cells were grown in succinate, fumarate, malate, pyruvate or acetate as the sole carbon source. Further comparisons of Av-NifA-E356K activity were performed comparing sucrose and acetate as carbon sources given that the growth penalty difference between the DJ and EK strains was significantly alleviated in acetate (S5 Fig).

When wild type *A. vinelandii* was subjected to a carbon shift from sucrose to acetate, a significant reduction in both *nifH* expression (Fig 3A) and nitrogenase activity (Fig 3B) was observed and as expected, both activities were repressed in the presence of ammonium (+N). However, the ability of the EK strain to escape regulation by fixed nitrogen (+N) was severely compromised when grown on acetate (Fig 3D and 3E). These results demonstrate that NifA-E356K responds to carbon status regulation *in vivo*, as anticipated from the *in vitro* characterization experiments [15,22]. Since 2-oxoglutarate is required to activate NifA only when NifL is present, we examined nitrogen and carbon regulation in the previously characterised strain AZBB163 in which *nifL* is disrupted by a kanamycin resistance cassette. [20]. In this case, in contrast to NifA-E356K, nitrogenase activity was constitutive and not strongly influenced by the carbon source (S6B Fig). However, this strain exhibited unexpected patterns of *nifH* expression when grown on sucrose that did not correlate with nitrogenase activity

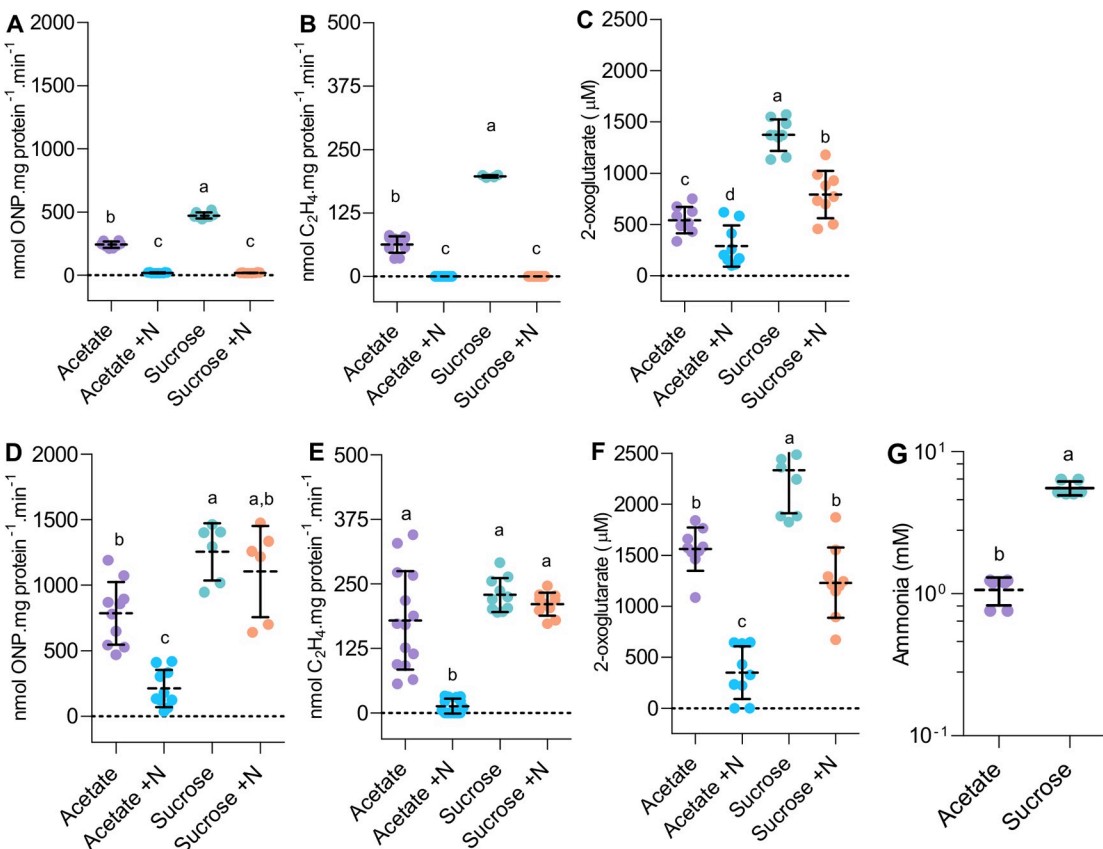

**Fig 3. Carbon status regulation of nitrogenase expression and ammonia excretion in *A. vinelandii*.** (A) Nitrogenase expression reported from a *nifH::lacZ* fusion, (B) nitrogenase activity and (C) internal 2-oxoglutarate levels are compared in the wild type (DJ) in acetate and sucrose in the absence (-N) or presence (+N) of 20 mM ammonium chloride. (D-F) The same comparisons as in (A-C) were done for the *nifA-E356K* strain (EK). For experiments in (A-F), strains were cultured to an O.D$_{600nm}$ between 0.2–0.3 in 30 mM acetate or 60 mM sucrose. To facilitate direct comparison of nitrogenase expression and activity, strain DJHZ was used in panels (A-B) whereas strain EKHZ was used in panels (D-E). These strains are isogenic to DJ and EK, respectively, except that they encode a *nifH::lacZ* fusion in the *algU* locus, a neutral site in the *A. vinelandii* genome (S1 Table). (G) Ammonia levels detected in the *nifA-E356K* (EK) strain when grown in either acetate or sucrose until cultures reached stationary phase. Plots followed by different letters are statistically different according to ANOVA with post-hoc Tukey's HSD or a paired t-test in (G).

(S6A Fig), potentially because *nifA* expression is not driven by the native *nifL* promoter in strain AZBB163 [20]. Taken together, these results demonstrate that nitrogenase expression and activity is suppressed in the EK strain when grown under nitrogen excess conditions with acetate as the sole carbon source. Since the NifA-E356K protein is unable to escape nitrogen regulation mediated by NifL and GlnK when 2-oxoglutarate is limiting *in vitro*, this metabolite is likely to provide the physiological signal that triggers the carbon source response. Quantification of internal 2-oxoglutarate levels in strains grown on the different carbon sources (Fig 3C and 3F) supports previous evidence that the levels of this metabolite are sensitive to the carbon and nitrogen supply [16,33]. In the wild type strain (DJ), 2-oxoglutarate levels dropped significantly in acetate compared to sucrose and a further decrease was observed when excess fixed nitrogen was present regardless of the type of carbon source (Fig 3C). 2-oxoglutarate levels in the *nifA-E356K* strain were generally higher than in the wild type, but were influenced in a similar manner in relation to carbon and nitrogen source availability (Fig 3F). Overall, the fluctuations in the 2-oxoglutarate levels correlated well with nitrogenase activity and expression for both the wild type (compare Fig 3A, 3B and 3C) and the *nifA-E356K* mutant (compare

Fig 3D, 3E and 3F), reinforcing the importance of carbon signalling in the regulation of nitrogen fixation. Notably, when the 2-oxoglutarate level decreased below 350 μM in the *nifA-E356K* strain (Fig 3F, acetate +N condition) nitrogen regulation was less effectively bypassed *in vivo* (Fig 3D and 3E), commensurate with previous *in vitro* biochemical data [15,34]. Consequently, lower levels of ammonia excretion were detectable when the *nifA-E356K* strain was grown on acetate (Fig 3G). In accordance with this, the growth rate penalty observed in the EK (*nifA-E356K*) strain in the presence of sucrose was significantly reduced when acetate was the carbon source (S5 Fig).

## NifA-E356K is a prototype for engineering conditional ammonia excretion in diazotrophic Proteobacteria

Although regulation by NifL is not the most prevalent mechanism for controlling NifA activity in bacteria, the *nifL-nifA* operon is widely distributed in Proteobacteria [3] and the glutamate residue at position 356 in *A. vinelandii* NifA is highly conserved (Fig 4 panels A and B). We

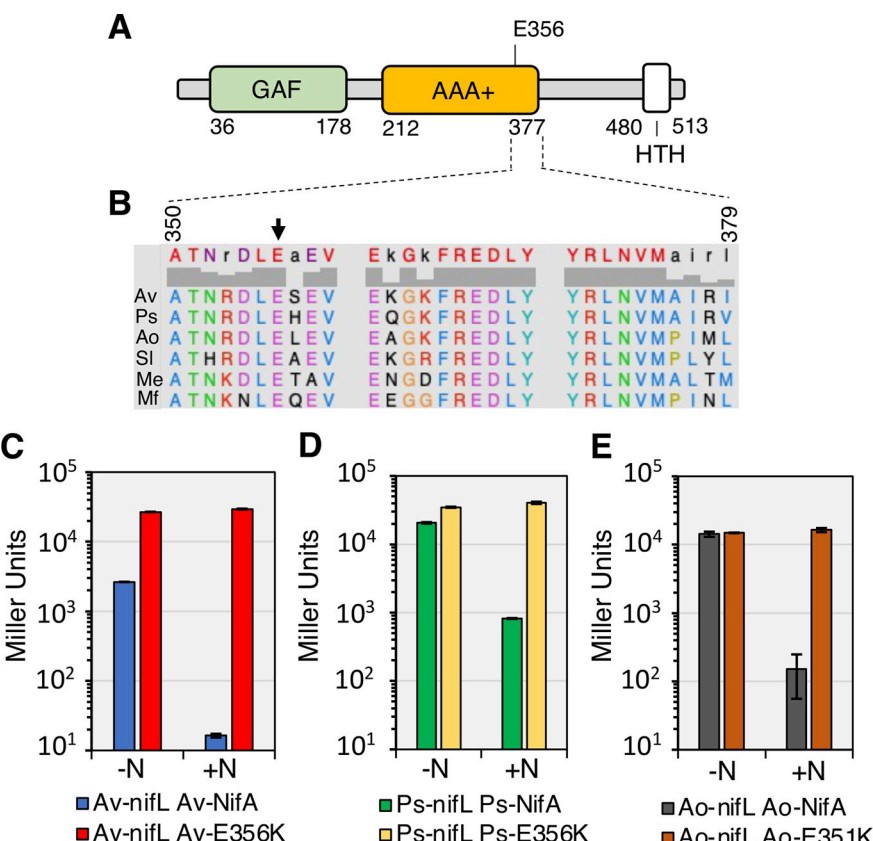

**Fig 4. Reciprocal amino acid changes (related to *nifAE356K* in *A. vinelandii*) yield constitutively active NifA in Proteobacteria.** (A) Diagram of the *A. vinelandii* NifA domains. (B) Alignment of residues close to E356 (black arrow) in the central AAA+ domain of NifA proteins regulated by NifL. Sequence numbers refer to *A. vinelandii* NifA. Sequences used in the alignment are Av: *A. vinelandii* DJ, Ps: *Pseudomonas stutzeri* A1501 (Gammaproteobacteria), Ao: *Azoarcus olearius* DQS4, Sl: *Sideroxydans lithotrophicus* ES-1 (Betaproteobacteria), Me: *Martelella endophytica* YC6887 (Alphaproteobacteria) and Mf: *Mariprofundus ferrooxydans* M34 (Zetaproteobacteria). Panels C to E show β-galactosidase activities in the *E. coli* ET8000 chassis resulting from activation of a *nifH::lacZ* fusion (plasmid pRT22) by wild type and variant NifL-NifA systems from three different diazotrophs. Plasmids used to express NifL-NifA variants are as follows. (C) pPR34: Av-NifL-NifA, pPMA: Av-NifL-NifA-E356K; (D) pMB1804: Ps-NifL-NifA, pMB1805: Ps-NifL-NifA-E356K; (E) pMB1806: Ao-NifL-NifA, pMB1807: Ao-NifL-NifA-E351K. The assays were performed in NFDM media supplemented with 2% glucose in either nitrogen-limiting (200 μg/ml of casein hydrolysate, -N) or nitrogen excess (7.56 mM ammonium sulphate, +N) conditions.

therefore sought to evaluate if introduction of the reciprocal amino acid substitution in NifA proteins from other Proteobacteria, would yield the same regulation profile as in *A. vinelandii*. Using a previously established two-plasmid system to study the *A. vinelandii* NifL-NifA system in an *E. coli* background [11] we evaluated the activity of NifA variants from *Pseudomonas stutzeri* A1501 [23,24] and *Azoarcus olearius* DQS-4 [35,36]. Both of these species are thought to be well adapted for the endophytic lifestyle and therefore are attractive model organisms for engineering ammonia excretion to benefit plant growth. The activities of wild type *P. stutzeri* NifA and *A. olearius* NifA in the presence of their corresponding NifL partners were higher than wild type *A. vinelandii* NifA under nitrogen-limiting conditions (-N) but as expected, were strongly inhibited in the presence of fixed nitrogen (+N) (Fig 4, panels C-E). In contrast, the reciprocal NifA-E356K substitutions in *P. stutzeri* NifA (Ps-NifA-E356K) and *A. olearius* NifA (Ao-NifA-E351K) gave rise to constitutive activation of the *nifH* promoter in nitrogen replete (+N) conditions in *E. coli* (Fig 4 panels D-E, respectively).

To examine the properties of the Ps-NifA-E356K substitution in its endophytic host, we introduced the corresponding *nifA* mutation into the chromosome of *P. stutzeri* A1501. However, contrary to *A. vinelandii*, this single mutation (in strain Ps-EK) did not result in constitutive *nifH* transcription (S7 Fig), presumably because expression of the *nifLA* operon itself is regulated by nitrogen availability in *P. stutzeri* [24,37]. In order to remove this second layer of nitrogen regulation, we replaced the native *P.stutzeri nifL* promoter with the *A. vinelandii nifL* promoter. Although this replacement (in the strain Ps_nifLA^C) suppressed nitrogen control of *nifA* transcription as anticipated, constitutive *nifH* transcription was not observed, confirming that the *P. stutzeri* wild-type NifL-NifA system remains responsive to nitrogen regulation when expressed constitutively (S7 Fig). To examine the intrinsic ability of Ps-NifA-E356K to escape nitrogen control, we combined the *A. vinelandii nifL* promoter replacement with the *nifA-E356K* mutation in *P. stutzeri* (Fig 5A). Although this strain (Ps-EK^C) expressed relatively low levels of *nifLA* transcripts under diazotrophic (-N) conditions (S7F–S7G Fig, direct correlation between the levels of *nifLA* and *nifH* transcripts was observed in excess nitrogen (+N) conditions, confirming that the E356K substitution enables Ps-NifA to escape nitrogen regulation mediated by NifL and GlnK in *P. stutzeri* (S7F–S7H Fig).

The Ps-EK^C strain was able to activate *nifH* transcription on a variety of carbon sources when grown under nitrogen excess conditions (+N) with lactate yielding the highest level of *nifH* activation followed by glucose, malate and glycerol (Fig 5B). This carbon source-dependent activation of *nifH* transcription by Ps-NifA-E356K correlated directly with the level of nitrogenase activity in each condition (compare Fig 5B and 5C). As anticipated from the relatively high level of *nifH* transcription and nitrogenase activity conferred by growth on lactate, ammonia excretion was only observed when the Ps-EK^C strain was grown on this carbon source (Fig 5D). Finally, we observed no growth penalty for the Ps-EK^C strain when grown on complex media (LB) or in minimal media supplemented with glucose, malate, lactate or glycerol as carbon sources (S8 Fig), which implies that the relatively moderate activation of *nif* gene transcription in the Ps-EK^C strain, allows carbon regulated ammonia excretion without severe impacts to bacterial fitness. Altogether, these observations suggest that introducing the reciprocal E356K substitution into NifA proteins from diazotrophic Proteobacteria, may be broadly applicable for engineering new bacterial strains with carbon-controlled excretion of ammonia.

## Discussion

In order to cope with the energetic cost of biological nitrogen fixation, diazotrophic bacteria require sophisticated signal transduction mechanisms ensuring efficient adaption to changing

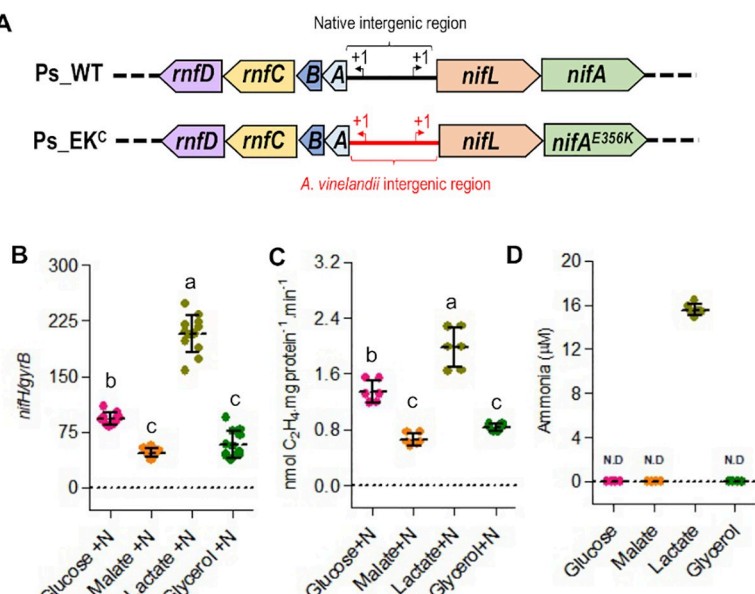

**Fig 5. Ps-NifA-E356K protein is able to escape NifL inhibition in *P. stutzeri* leading to carbon-dependent ammonia excretion. The** (A) Diagram depicting the modifications in the *P. stutzeri nifA-E356K* strain (Ps_EK$^c$) compared to wild type *P. stutzeri* (Ps_WT). Drawings are not to scale. (B) Levels of *nifH* transcripts in the *P. stutzeri* Ps_EK$^c$ strain in the presence of 5 mM NH$_4$Cl (+N) under 4% O$_2$. (C) Nitrogenase activity in the *P. stutzeri* strain (Ps_EK$^c$) in the presence of 5 mM NH$_4$Cl (+N) under 8% O$_2$. (D) Comparison of ammonia excretion profiles in different carbon sources. In all cases, the Ps_EK$^c$ strain was grown in UMS-PS supplemented with 30 mM glucose, 45 mM malate, 60 mM lactate or 60 mM glycerol (to provide balanced carbon equivalents). In (D) cells (O.D$_{600\ nm}$ = 0.2) were incubated at an initial O$_2$ concentration of 8% for 48 hours before ammonia quantification. N.D: not detected. Plots followed by different letters are statistically different according to ANOVA with post-hoc Tukey's HSD.

conditions whilst successfully competing in the environment. Achieving an appropriate balance between carbon and nitrogen metabolism is particularly challenging for organisms that fix nitrogen, requiring diversion of ATP and reducing equivalents from central metabolism to ensure nitrogenase catalytic rates that meet the nitrogen demands required for growth. Hence the ability to sense carbon availability in addition to the nitrogen status, is paramount to resource allocation and to resolve conflicting metabolic demands.

The physiological signal for carbon status control is most likely to be 2-oxoglutarate given the correlation observed here between the level of this metabolite with nitrogen regulation *in vivo*, together with our previous biochemical demonstration of the importance of this ligand in NifL-NifA regulation [14,15]. We propose that this additional level of metabolite regulation provides a mechanism to integrate signals of the carbon and nitrogen status to ensure that sufficient carbon resources are available to support diazotrophy. The *in vitro* data indicate that when 2-oxoglutarate is limiting, NifL, forms a binary complex with NifA, which inhibits its activity, even under nitrogen-limiting conditions when GlnK is fully uridylylated and unable to interact with NifL (Fig 6A). However, when sufficient levels of 2-oxoglutarate are available (Fig 6B) the NifL-NifA complex dissociates, enabling NifA to activate *nif* transcription [2,4,14,15]. Upon a switch to excess nitrogen conditions, GlnK becomes de-uridylylated allowing the formation of a ternary GlnK-NifL-NifA complex that inactivates NifA irrespective of the level of 2-oxoglutarate (Fig 6C and 6D). Hence in the wild-type NifL-NifA system, the nitrogen status signal overrides the metabolic signal of the carbon status, when excess fixed nitrogen is available. In contrast, in the variant NifA-E356K protein studied here, the integration between nitrogen and carbon control is disrupted. Although this substitution in the AAA

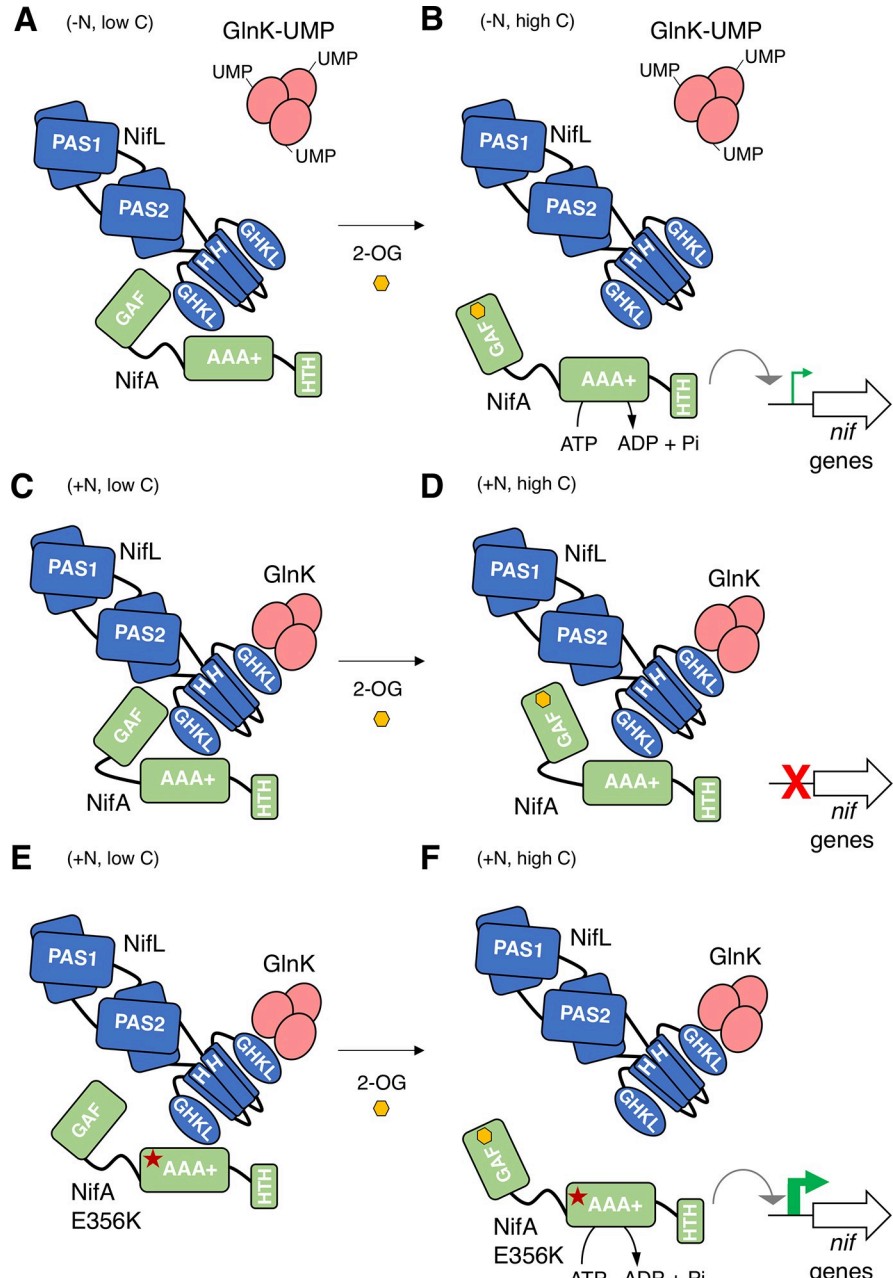

**Fig 6. Model for 2-oxoglutarate regulation of NifA activity based on genetic and biochemical experiments.** (A) When 2-oxoglutarate levels are low, NifL can inhibit NifA even under nitrogen-limiting conditions (-N, low C) when GlnK is uridylylated (GlnK-UMP) and unable to interact with NifL. (B) Binding of 2-oxoglutarate (2-OG, yellow hexagon) to the GAF domain of NifA under carbon sufficient conditions (-N, high C), disrupts the binary NifL-NifA interaction, thus activating NifA. (C) Under nitrogen excess conditions (+N, low C), non-covalently modified GlnK, interacts with the GHKL domain of NifL, stimulating the formation of a ternary complex between GlnK, NifL and NifA that inhibits NifA activity. (D) The GlnK-NifL-NifA ternary complex is stable under nitrogen excess conditions even if the GAF domain in NifA is saturated with 2-OG (+N, high C). (E) The E356K substitution in the AAA + domain of NifA (red star), perturbs the interaction with NifL in the absence of the non-modified form of GlnK. However, under nitrogen excess conditions, the GlnK-NifL-NifA-E356K ternary complex is formed when 2-OG is limiting, as a consequence of carbon limitation (+N, low C). (F) Upon a switch to a preferred carbon source, when 2-OG levels are sufficient (+N, high C), conformational changes triggered by 2-OG binding to the GAF domain disrupt the ternary complex, activating NifA-E356K under conditions of nitrogen excess.

+ domain of NifA (red star, Fig 6E) perturbs the interaction with NifL, the GlnK-NifL-NifA-E356K ternary complex still forms if 2-oxoglutarate is limiting (Fig 6E). In contrast, when 2-oxoglutarate levels are sufficient, conformational changes triggered by its binding to the GAF domain disrupt the ternary complex, enabling NifA-E356K to be active in the presence of excess fixed nitrogen (Fig 6F). Therefore, although the E356K substitution escapes nitrogen control, conferred by resistance to the GlnK bound form of NifL, this is contingent upon the binding of 2-oxoglutarate to the GAF domain of this variant protein. The response of the NifL-NifA system to 2-oxoglutarate thus emphasises the key role of this metabolite as a master signalling molecule [16]. Consequently, the ability of the NifA-E356K variant to bypass nitrogen regulation *in vivo* in both *A. vinelandii* and *P. stutzeri* is dependent on the carbon status. The crucial role of carbon-mediated signalling in the regulation of nitrogen fixation was evident from reduced *nifH* transcripts and activity of nitrogenase when the *A. vinelandii nifA-E356K* strain was cultured under nitrogen excess conditions with acetate as sole carbon source, which correlated with a significant decrease in the level of 2-oxoglutarate and a 6-fold reduction in ammonium excretion compared with sucrose as carbon source. Similarly, in *P. stutzeri* where lactate appears to be a preferred carbon source to support nitrogen fixation in comparison to glucose, the *nifA-E356K* mutant exhibited the highest level of *nifH* transcripts and nitrogenase activity under nitrogen excess conditions when lactate was provided as a carbon source. Not surprisingly, amongst the carbon sources tested, the *P. stutzeri* EK$^C$ strain only excreted ammonia when provided with lactate under our experimental conditions. The capacity for runaway expression of *nif* genes, constitutive nitrogenase activity and ammonia excretion is therefore dependent on the nature of the carbon source. Our studies with *A. vinelandii* and *P. stutzeri* therefore demonstrate the potential to exploit the intrinsic carbon-sensing mechanism of the NifL-NifA system to provide conditional release of fixed nitrogen and hence alleviate the fitness penalty associated with constitutive expression of nitrogenase.

Bypassing nitrogen regulation of the NifL-NifA system to activate constitutive expression of nitrogenase would not by itself be anticipated to promote ammonia release, if the excess ammonia can be assimilated by the GS-GOGAT pathway. We have demonstrated that overexpression of nitrogenase in the *A. vinelandii nifA-E356K* strain leads to feedback regulation of GS activity via co-valent modification by the adenylyl transferase activity of GlnE. This reduction of nitrogen assimilation via post-translational modification of glutamine synthetase is a key factor in enabling ammonia excretion, which is not observed in the *nifA-E356K* strain when the *glnE* gene is deleted. Hence, in *A. vinelandii*, ammonia excretion is also dependent on the native feedback regulation of GS activity, exacerbated by higher rates of nitrogen fixation in the *nifA-E356K* strain when grown on a carbon source that sustains high levels of 2-oxoglutarate under excess nitrogen conditions. This uncoupling of nitrogen fixation from ammonium assimilation is somewhat analogous to what is observed in differentiated nitrogen-fixing bacteroids in the legume-rhizobium symbiosis, where the flux through the ammonia assimilation pathway is severely restricted to enable release of most of the nitrogen fixed by the symbiont [38,39]. Analogous strategies to decrease the activity of GS in non-symbiotic bacteria have resulted in ammonia excretion [40–42], but to date have not been combined with mutations that express high levels of nitrogenase on a conditional basis as deployed here.

Since introduction of the reciprocal E356K substitution into NifA proteins from other members of the Proteobacteria also results in nitrogen-insensitive activators when analysed either in a heterologous chassis, *E. coli* (Fig 4), or in *P. stutzeri* (Fig 5), this strategy may allow generation of new diazotrophic strains with conditional excretion of ammonia. The carbon responsive control mechanism could present an opportunity for activation of ammonium excretion contingent upon carbon sources provided by root exudates of crops, a feature highly desirable in the engineering of a synthetic symbiosis [43,44]. However, given that the

composition of root exudates is dynamic and difficult to define, engineering of the partner plant to provide a favourable carbon source may also be required. In addition, as further regulatory complexities are associated with fine-tuning nitrogen regulation in diverse Proteobacteria, other manipulations to disrupt transcriptional control of NifL-NifA expression itself or the coupling between nitrogen fixation and assimilation may be needed to achieve ammonia excretion [3,45]. One such example explored in this study was the need to remove native nitrogen regulation from the *P. stutzeri* A1501 *nifLA* promoter. Serendipitously, we demonstrated that providing relatively low levels of the *nifA-E356K* transcripts in *P. stutzeri* generated a strain with carbon-regulated ammonia excretion without the severe growth penalties observed in *A. vinelandii*. Furthermore, we demonstrated that the levels of ammonia excretion are directly correlated with specific nitrogenase activity rates in both organisms analysed. Under optimal conditions of carbon and oxygen supply, the *A. vinelandii nifA-E356K* mutant sustained a very high rate of nitrogenase activity (200–300 nmol $C_2H_4$.mg protein$^{-1}$.min$^{-1}$) under nitrogen excess conditions. On the other hand, in *P. stutzeri* the levels of nitrogenase activity conferred by *nifA-E356K* were at least 100-fold lower under our experimental conditions. Hence, *A. vinelandii* excreted millimolar levels of ammonia in contrast to the micromolar levels observed in *P. stutzeri.* As the ammonium excreting strain from *P. stutzeri* was not subject to the same growth penalty observed in the *A. vinelandii* counterpart, we anticipate that these studies will guide future efforts to define more precise trade-offs to engineer nitrogen releasing strains that do not have a competitive disadvantage in the rhizosphere. Moreover, the addition of multi-layered regulatory control of ammonia excretion by expressing activator variants under the control of promoters that respond to specific signalling molecules exchanged between the plant and the bacteria, may deliver the required level of specificity for the establishment of an efficient synthetic symbiosis [46,47].

## Materials and methods

### Bacterial strains and growth conditions

The bacterial strains used in this study are listed in S1 Table. For routine procedures *E. coli* strains were grown at 37˚C in LB medium [48] or in NFDM [49] for β-galactosidase activity assays. Media for *E. coli* ST18 [50] was supplemented with 50 μg/mL of ALA (5-aminolevulinic acid) unless counter-selection was required. *A. vinelandii* was grown at 30˚C and 250 rpm in NIL medium (containing 0.2 g/L $MgCl_2$, 90 mg/L $CaCl_2$, 0.8 g/L $KH_2PO_4$, 0.2 g/L $K_2HPO_4$, 14 mg/L $Na_2SO_4$, 120 mg/L $Fe_2(SO_4)_3$ and 2.4 mg/L $Na_2MoO_4$) supplemented with either 2% sucrose (approximately 60 mM) as described previously [22,51] unless stated otherwise. Alternatively, as the NIL medium was prone to precipitation hampering efforts to automate the measurement of growth rate parameters, a novel minimal medium, hereafter named MBB (*M*inimal *B*acterial *B*roth), was used for automated growth experiments on a plate reader. MBB contained 0.6 g/L $K_2HPO4$, 0.4 g/L $KH_2PO4$, 1.1 g/L NaCl, 0.4 g/L $MgSO_4$, 20 mg/L $CaCl_2.2H_2O$, 10 mg/L $MnSO_4.H_2O$, 65.6 mg/L Fe(III)-EDTA, 2 mg/L $Na_2MoO_4.2H_2O$ and 2% (20 g/L) sucrose. No apparent differences in the growth rate, ammonium excretion or diazotrophy (ability to grow on atmospheric $N_2$), were observed comparing NIL and MBB media when 2% sucrose was used as carbon source. Therefore, implementation of MBB allowed automation of growth parameter measurements, without detrimental effects to physiological traits of interest. *P. stutzeri* was grown at 30˚C in LB for routine pre-inocula and for conjugations. For experiments requiring defined media such as measurement of nitrogenase activity, *P. stutzeri* was grown in a modified version of the UMS medium [52], hereafter named UMS-PS. UMS-PS was obtained by supplementing the UMS medium with 1:1000 dilution of vitamin solution (1 g/L thiamine hydrochloride, 2 g/L D-pantothenic acid, 1 g/L nicotinic acid, and 0.1

g/L biotin) and 50 mL/L of Kalininskaya phosphate buffer (KP buffer–$K_2HPO_4$ 33.4 g/L, $KH_2PO_4$ 17.4 g/L [53]), which significantly helped to reduce flocculation. The carbon source used for *P. stutzeri* was 2% glucose unless stated otherwise. 5–10 mM of $NH_4Cl$ was added to UMS-PS for non-diazotrophic growth as indicated. Antibiotics were used as follows: carbenicillin 50 μg/mL (*E. coli*), chloramphenicol 15 μg/mL (*E. coli*), tetracycline 5 μg/mL (*E. coli, A. vinelandii* and *P. stutzeri*), kanamycin 50 μg/mL (*E. coli* and *P. stutzeri*) and 1–3 μg/mL (*A. vinelandii*), trimethoprim 15 μg/mL (*E. coli*) and 90 μg/mL (*A. vinelandii*).

## Preparation of *A. vinelandii* competent cells and transformation

Competent cells of *A. vinelandii* were obtained in molybdate and iron depleted competence medium essentially as in [51], except that the medium was amended with 7 mM of $MgCl_2$. Strains streaked for single colonies in competence medium plates were incubated at 30˚C until a green fluorescent siderophore was produced (5–7 days), indicative of competence [51,54]. Subsequently, a loopful of cells was resuspended in 500 μL of P-buffer (4.6 mM $K_2HPO_4$,1.5 mM $KH_2PO_4$) and then mixed with 250–500 ng of a linear DNA fragment carrying the desired mutation. This mixture was then spotted onto the centre of a competence medium plate and further incubated overnight at 30˚C. On the following day, cells were scraped off the plates and resuspended into 1 mL of P-buffer as above. A serial dilution was then spread onto NIL medium supplemented with 25 mM ammonium acetate with antibiotics to obtain single colonies. In the case of the *nifA-E356K* mutant, recombinant colony selection was achieved by recovery of diazotrophy (ability to grow on atmospheric $N_2$) using competent cells from a *nifA* deletion strain (DJA), without antibiotic selection, to generate the strain EK. Given that *A. vinelandii* can accumulate up to 80 copies of its chromosome under certain conditions [51,55] newly recombinant colonies were exhaustively streaked on selective media (20 times or more) to ensure efficient chromosome segregation and homogeneity of the mutant genotype.

## Recombinant DNA work

General molecular biology techniques were performed according to established protocols [48]. Enzymatic isothermal assembly [56] was performed with the NEBuilder HiFi DNA Assembly Master Mix (NEB #E2621). Site-direct mutagenesis by overlapping PCR was performed as described previously [57]. High-fidelity DNA polymerase and restriction enzymes were provided by New England Biolabs. DNA purification was performed using commercially available kits provided by Macherey-Nagel. Sanger DNA sequencing and oligonucleotide synthesis was conducted by Eurofins MWG Operon.

## Construction of *A. vinelandii* mutants

To construct the *A. vinelandii nifA-E356K* mutant strain (EK), a fragment of 1569 pb corresponding to the *A. vinelandii nifA* gene carrying discrete base pair changes, GAA->AAG (yielding the E356K substitution), was amplified by PCR from the plasmid pAAS1544 (S2 Table) with primers ASS-3 and ASS-55 (S3 Table). The above base change also introduced a recognition site for the restriction enzyme, AcuI, which assisted genotypic screening. This PCR fragment was then directly transformed into a *nifA* deletion background and recombinant colonies were selected on the basis of diazotrophy recovery as described above. To construct mutants in the structural nitrogenase gene (*nifH*), a fragment of 1874 pb corresponding to the *nifHD* region (position 136301–138174) was amplified with primers 2–45 and 2–46 and cloned as a BamHI/HindIII fragment into pBlueScript II KS +, yielding pMB1724. Subsequently, the *tetA* gene (tetracycline resistance) was PCR amplified from pALMAR3 (primers 2–43 and 2–44) and cloned into the BglII and EcoRI sites of pMB1724 to yield pMB1725. This

plasmid (Δ*nifH*::*tetA*), was linearized with ScaI and transformed, as described above, into *A. vinelandii* wild type (DJ) and *nifA-E356K* (EK) to generate the strains DJH and EKH, respectively. Construction of the *glnE* gene deletion strain (EKΔE), which carries a trimethoprim (*tmp*) resistance marker was performed by transformation of a linear PCR product obtained with primers M13F(-47) and M13R(-48) from the plasmid pMB1840. This plasmid, was obtained by isothermal assembly of fragments upstream (906 bp, position 4547284–4548189, primers 7-28/7-29) and downstream (752 bp, position 4543778–45444529, primers 7-32/7-33) of *glnE*, with a 631 bp fragment, obtained from pUC18T-mini-Tn7T-Tp with primers 7-30/7-31, encoding the *tmp* resistance gene, into pk18mobsacBKm linearized with SmaI. *A. vinelandii* reporter strains were obtained by insertion of a *nifH*::*lacZ* fusion into the *algU* locus. In *A. vinelandii* (DJ), the *algU* gene is naturally inactivated by an insertion sequence yielding the non-gummy phenotype [58,59]. We therefore anticipated that this was a convenient locus to insert a *nifH*::*lacZ* reporter whilst keeping the original *nifHDK* locus intact and avoiding problems that may arise from plasmid instability. This strategy facilitated comparison of both nitrogenase and NifA activities in a single *A.vinelandii* strain background. To obtain the *nifH*::*lacZ* reporter strains, the plasmid pMB1816 was linearized with ScaI and transformed into *A. vinelandii* as described above. pMB1816 was constructed by isothermal assembly of a 409 bp fragment corresponding to the *nifH* promoter (position 136401–136809, primers 5-49/5-50) with a 3057 bp (primers 5-46/5-464) and the *lacZ* gene derived from pRT22 into an *algU* integrative plasmid backbone. The latter was obtained by assembly of fragments upstream (795 bp, position 1329654–1330448) and downstream (813 bp, position 1332074–1332886) of the *algU* gene with an 1811 bp fragment corresponding to the *ori* and ampicillin resistance gene from pUC19 and a 631 bp fragment carrying the trimethoprim resistance gene.

## Construction of *P. stutzeri* mutants

Mutagenic suicide plasmids based on the pk18mobsacBKm vector were conjugated into *P. stutzeri* essentially as described [60] by mixing recipient and donor strains in two proportions (50:1 and 10:1), except that the *E. coli* strain ST18 [50] was used as donor and that the whole procedure was performed on LB-agar medium. The *E. coli* ST18 was counter-selected after biparental mating based on its auxotrophy to ALA (5-aminolevulinic acid). Selection of double crossovers was performed in LB-agar supplemented with 10% sucrose. The *P. stutzeri* strain Ps_nifLA^C, was obtained by replacing the native *rnf-nifLA* intergenic region by the reciprocal region from *A.vinelandii* using pMB2005. This plasmid was obtained by fusing the *rnf-nifLA* intergenic region (444 bp; primers 8-19/8-20) from *A. vinelandii* downstream to a fragment of the *P. stutzeri rnfAB* genes (1160 bp; primers 8-17/8-18) and upstream to a fragment of the *P. stutzeri nifL* gene (1638 bp; primers 8-21/8-22) using isothermal assembly into pk18mobsacBKm linearized with SmaI. To generate the *P. stutzeri* strain Ps_EK^C (Fig 5A) the plasmid pMB2006 was used to the generate the strain Ps_EK^C-*tetA*. Subsequently the *tetA* resistance cassette was recovered from the genome using pMB2007. The plasmid pMB2006 was obtained by fusing 847 bp downstream Ps-*nifA* (primers 8-2/8-1) with a 2364 bp fragment from the Ps-*nifLAE356K* (primers 8-3/8-11b; from pMB1805) by isothermal assembly into pk18mobsacBKm linearized with SmaI. To obtain pMB2007, pMB2006 was linearized by PCR with primers 8-13/8-14 (8878 bp) and subsequently fused to the *tetA* gene fragment (1349 bp; primers 8-15/8-16) by isothermal assembly.

## Construction of plasmids expressing NifL-NifA from other Proteobacteria

A plasmid, pPR34, for the expression of the *nifLA* operon from *A. vinelandii* has been previously constructed [11] allowing activation of a *nifH*::*lacZ* fusion in *E. coli* ET8000. To compare

the activity of NifL-NifA proteins from *A. olearius* DQS4 (Ao) and *P. stutzeri* A1501 (Ps) with the archetypal proteins from *A. vinelandii* (Av), while avoiding differences that may arise from plasmid copy number or protein expression levels, we constructed a series of plasmids derived from pPR34 to express *A. olearius* NifL-NifA and *P. stutzeri* NifL-NifA. The pPR34 backbone, excluding the *nifLA* genes from *A. vinelandii*, was amplified with primers 3–75 and 4–14 to generate a 2421 bp fragment. To allow isothermal enzymatic assembly, 15–25 bp overlaps to the linearized pPR34 backbone were introduced into the 5' and 3' ends of *nifL* and *nifA*, respectively, from *A. olearius* or *P.stutzeri*. The derived plasmids, pMB1806 (*A. olearius* NifL-NifA) and pMB1804 (*P. stutzeri* NifL-NifA) were the templates for the introduction of point mutations by overlapping PCR mutagenesis to generate plasmids pMB1807 (*A. olearius* *nifA-E351K*) and pMB1805 (*P. stutzeri* *nifA E356K*). All plasmids and primers are listed in S2 and S3 Tables respectively.

## Quantification of ammonia

Ammonia from culture supernatants was quantified by the indophenol-blue method modified from [61,62]. 750 μL of sample (usually 50–350 μL of supernatant diluted to 750 μL in $H_2O$), or the calibration curve standards, were mixed with 150 μL of sodium phenate (0.25 M phenol, 0.3 M NaOH), 225 μL of 0.66 mM sodium nitroprusside dihydrate and 225 μL of 1–1.5% sodium hypochlorite, in this order. The final composition of the reaction mix was approximately: 28 mM phenol, 33 mM NaOH, 25 mM sodium hypochlorite and 0.11 mM sodium nitroprusside. Samples were incubated for approximately 40 minutes at room temperature and the resulting indophenol-blue quantified at 625 nm. The calibration curve used known concentrations of $NH_4Cl$ ranging from 0 to 0.3 mM (in 0.05 increments) from a 1 mM standard.

## β-galactosidase activity

β-galactosidase activity assays were performed as described previously [63] and reported in Miller Units for *E. coli* or as specific activity units (nmol ONP.mg protein$^{-1}$.min$^{-1}$) for *A. vinelandii*. Growth conditions for assays in *E. coli* ET8000 were as previously established [11,21,22], except that 6 mL screw capped bijou universals were filled to the brim to enable anaerobic conditions [64]. For assays involving *A. vinelandii* strains, cultures were prepared as for the nitrogenase activity measurements as described below. Specific β-galactosidase activity was calculated using an ONP (o-nitrophenol–Sigma# N19702) calibration curve prepared in the reaction buffer containing the same amount of $Na_2CO_3$ used to stop β-galactosidase development, as addition of $Na_2CO_3$ intensifies ONP colour development [63]. Whole cell protein concentration was determined by the Bradford method [65] after overnight cell lysis in 0.1 mM NaOH.

## Nitrogenase activity

*In vivo* nitrogenase activity in *A. vinelandii* and *P. stutzeri* was measured by the acetylene reduction assay [66,67] in cultures growing in liquid media as indicated below. After incubation of cells with acetylene for 0.5- to 1-hour (*A. vinelandii*) or 4–18 hours (*P. stutzeri*), ethylene ($C_2H_4$) was quantified using a Perkin Elmer Clarus 480 gas chromatograph equipped with a HayeSep N (80–100 MESH) column. The injector and oven temperatures were kept at 100˚C, while the FID detector was set at 150˚C. The carrier gas (nitrogen) flow was set at 8–10 mL/min. Nitrogenase activity is reported as nmol of $C_2H_4$. mg protein$^{-1}$.min$^{-1}$. The ethylene calibration curve was prepared from chemical decomposition of ethephon (Sigma #C0143) in a 10 mM $Na_2HPO_4$ pH = 10.7 as described previously [68]. As *A. vinelandii* is able to fix

nitrogen in air (21% $O_2$) 20 mL cultures were grown in NIL media as above in 100 mL conical flasks in an open atmosphere at 250 rpm until the desired $O.D_{600nm}$ was reached. Immediately before the acetylene reduction assay, the flask was stoppered with rubber septa (Suba-Seal n˚ 37), 10% acetylene injected, and the ethylene formed analysed after 0.5- to 1-hour of incubation. Given the microaerobic lifestyle of *P. stutzeri*, the flasks were prepared at defined initial oxygen concentrations of 4% or 8% in the gas phase of the flasks, based upon an $O_2$ titration to define appropriate conditions for rapid nitrogenase de-repression or ammonia excretion, respectively (S9 Fig). 50 mL of cells pre-cultured in LB overnight were collected by centrifugation and resuspended in 10–15 mL of UMS basal medium without carbon source to wash away excess nitrogen. 20 mL of bacterial suspension at an initial $O.D_{600nm}$ of 0.2–0.4 (aiming for a final $O.D_{600nm}$ of 0.55–0.6 at the end of the incubation period) was prepared in UMS-PS medium, transferred to 100 mL conical flasks and stoppered with the rubber septa. The flasks were flushed with nitrogen for 20–25 min prior to adjusting the oxygen concentration by injecting back a defined amount of air into the flask. After oxygen concentration adjustment, the flasks were incubated for 4 hours at 120 rpm and 30˚C and then 10% acetylene was injected. The ethylene formed was analysed after 4–18 hours.

## Glutamine synthetase activity

Glutamine synthetase biosynthetic (GSB) and transferase (GST) activities were determined using previously described protocols [29,31] with minor modifications. Prior to the assay, 20 mL of cells grown under the conditions described earlier, were quenched by addition of 2 mL of CTAB (1 mg/mL) under constant agitation (250 rpm) for 3 minutes. Cells were then immediately collected by centrifugation at 4˚C, washed once in 20 mL of 1% KCl, collected by centrifugation once more, and finally resuspended to a final volume of 1 mL in 1% KCl. The resulting cell extract was kept on ice, and immediately used for the activity assays. For the biosynthetic activity assay (GSB), 40 μL of cell extract was added to 400 μL of GSB assay mix (234 mM imidazole hydrochloride pH 7.4, 58.6 mM hydroxylamine hydrochloride, 70.4 mM magnesium chloride hexahydrate, 209 mM L-sodium glutamate and 117.3 μg/mL of CTAB) and equilibrated at 37˚C for 5 minutes. The reaction was started by the addition of 60 μL 0.2 M ATP. For the transferase activity assay (GST), 50 μL of cell extract was added to 400 μL of GST assay mix (168.6 mM imidazole hydrochloride pH 7.15, 22.2 mM hydroxylamine hydrochloride, 0.34 mM manganese chloride, 31.5 mM sodium arsenate pH 7.15, 0.44 mM ADP and 111 μg/mL of CTAB) and equilibrated at 37˚C for 5 minutes. The reaction was started by the addition of 50 μL 0.2 M L-glutamine. Both reactions were incubated at 37˚C for 30–40 minutes and stopped by the addition of 1 mL of the stop mix (55 g/L of $FeCl_3.6H_2O$, 20 g/L of trichloroacetic acid, 21 mL/L of concentrated HCl). The product of both reactions, L-Glutamyl-γ-Hydroxamate (LGH), was quantified at 540 nm. The activity is reported as nmol LGH.mg protein$^{-1}$.min$^{-1}$.

## Fluorometric quantification of 2-oxoglutarate

Prior to metabolite extraction, 40 mL of cells were rapidly vacuum filtered through a 0.22 μM cellulose acetate filter coupled to a 50 mL falcon tube (Corning # CLS430320). The retained cell biomass was recovered from the filter in 2 mL of 0.3 M $HClO_4$. After recovery, 1.7 mL of the suspension was rapidly transferred to a chilled 2 mL tube, vigorously vortexed and centrifuged (17000 x g, 4˚C, 5 min) to remove cell debris. Then, 1.5 mL of the supernatant was transferred to a new tube and neutralized with 230 μL of 2M $K_2CO_3$ followed by a 5 min incubation on ice. To precipitate the excess $KClO_4$ formed, the extract was centrifuged (17000 x g, 4˚C, 5 min) and the supernatant carefully transferred to a fresh tube avoiding touching the white

precipitate ($KClO_4$). 1–2 µL of the neutralized extract was used to estimate the pH (7.5–8.5) using pH strips. The volume of 2M $K_2CO_3$ used was optimized to achieve complete neutralization. The extracts were then stored at -80°C or used immediately for fluorometric quantification of 2-oxoglutrate as described previously [34,69,70]. The detection assay was performed in 300 µL final volume and contained 100 mM imidazole-acetate buffer pH 7.0, 60 mM ammonium acetate, 10 µM NADH, 100 µM ADP and 0.075 µg of glutamate dehydrogenase (Sigma #G7882-100MG). Reactions were started by adding the enzyme and were incubated at 25°C for 20 minutes or until stabilization of the NADH fluorescence decay. The NADH fluorescence decay calibration curve was interpolated with 2-oxoglutarate standards ranging from 0.18 to 2.7 nmol prepared in a "mock" extraction solution obtained from $HClO_4$ neutralization with $K_2CO_3$ in the same fashion as in the metabolite extraction procedure above. 200 µL (out of 300 µL) of the reactions were used to measure the NADH decay ($\lambda_{ex}$: 340nm and $\lambda_{em}$: 460nm) using a 96-well black plate with clear bottom (Corning # CLS3603-48EA) and the BMG CLAR-IOstar plate reader. The internal metabolite concentration was calculated assuming that the cell-bound water content is approximately four times the bacterial dry weight [71].

## RNA purification and quantitative RT-PCR

Prior to RNA purification, *A. vinelandii* DJ (wild type) and the *nifA-E356K* mutant (EK) were submitted to an ammonium switch as described previously [72]. Briefly, cells growing under ammonium excess were collected by centrifugation and resuspended to an $O.D_{600\ nm}$ of 0.5 in fresh medium without added ammonium and subsequently incubated for 4 hours prior to RNA extraction. For *P. stutzeri*, the RNA was purified from cultures prepared exactly as described for the nitrogenase activity assay above. To ensure preservation of intracellular RNA, the cultures were immediately mixed with 1/5 of stop solution (5% Phenol saturated with 0.1 M citrate pH 4.3, 95% ethanol) [73] and then rapidly chilled on ice for 20 minutes. RNA was purified using the TRI Reagent (Sigma #T9424) following manufacturer instructions. Genomic DNA was removed using TURBO DNA-*free* DNAse (Ambion #AM1907) following the rigorous DNase treatment according to the manufacturer. cDNA synthesis was performed with SuperScript II Reverse Transcriptase (Invitrogen #18064014) using 0.1–1 µg of total RNA as recommended by the manufacturer. The resulting cDNA was diluted 5 to 20-fold (to fit the genomic DNA calibration curve) and 2 µL used as template in a 20 µL qPCR performed with the SensiFAST SYBR No-ROX Kit (#BIO-98005) reagent and the Bio-Rad CFX96 instrument. Absolute quantification of target genes (*nifH*, *nifL* and *nifA*) alongside the normalizing housekeeping gene (*gyrB*) was performed according to [74] using a $C_q$ calibration curve interpolated using serial dilutions of purified genomic DNA from *A. vinelandii* and *P. stutzeri*. Relative quantification was performed by the $2^{-\Delta Cq\Delta Cq}$ method according to [75]. Primers were designed and validated according to [76] ensuring comparable efficiencies and specificity as judged by the presence of a single peak in the melting curve. The primers used are listed in S3 Table.

## Supporting information

**S1 Fig. The *A. vinelandii nifAE356K* strain (EK) has a growth penalty that is dependent on overexpression of the nitrogenase structural genes (*nifHDK*).** (A) Strains were grown in MBB media supplemented with 2% sucrose either in the presence (+N) or absence (-N) of 25 mM ammonium acetate. The growth rates (B) and doubling times (C) in the exponential phase of growth are also shown. (D) Strains carrying a *nifH* insertion were grown only in the presence of 25 mM ammonium acetate, given that they are unable to grow diazotrophically. The growth rates (E) and doubling times (F) in the exponential phase of growth were

calculated from the data in (D). Cells were assayed for growth on a 24-well microplate (Greiner-Bio one #662160) using the Biotek EON plate reader as described in the Materials and Methods section. The absorbances recorded at 600 nm were corrected to a pathlength of 1 cm.
(TIF)

**S2 Fig. Levels of *glnA* transcripts in *A. vinelandii* strains grown under two nitrogen regimes.** The strains were grown in minimal media supplemented with 2% sucrose under diazotrophic conditions (-N) or in the presence of excess ammonium chloride (+N). Data is relative to the maximum level of detected transcripts in the wild type under diazotrophic conditions (DJ -N) estimated from absolute quantification. The data is representative from 2 independent RNA purifications performed in technical triplicate. Plots followed by different letters are statistically different according to ANOVA with post-hoc Tukey's HSD.
(TIF)

**S3 Fig. Ammonia excretion ceases in a strain unable to adenylylate glutamine synthetase.** Cells were spread on opposite sides of an agar plate containing NIL media with 2% sucrose but without fixed nitrogen. The *nifA* deletion (DJA) is unable to grow unless a source of fixed nitrogen is provided. When spread opposite to the wild type strain (DJ) the *nifA* deletion strain (DJA) was unable to grow (A). In contrast, DJA grew when spread opposite to the strain EK due to the diffusion of excreted ammonium (B). When the *glnE* gene is deleted in the *nifAE356K* background (strain EKΔE), ammonium excretion is impaired and so is growth of the *nifA* deletion strain (C).
(TIF)

**S4 Fig. Activation of *nif* gene expression by NifA-E356K in different carbon sources as reported by a *nifH::lacZ* fusion in *A. vinelandii* (strain EKHZ).** Cell suspensions (O.D$_{600}$ = 0.1) were spotted as 10 μL drops (triplicate) on minimal solid media plates supplemented with 10 mM NH$_4$Cl and the carbon sources indicated. After 18–36 hours incubation at 30°C, the grown bacterial biomass from triplicate spots were pooled and resuspended in 1 mL of PBS buffer. The β-galactosidase activity was performed using 100 μL of the PBS resuspended cells. Data is relative to the maximum level of detected activity in sucrose, calculated from specific β-galactosidase activity (1287.08 ±242.76 nmol ONP. mg protein$^{-1}$. min$^{-1}$). The data is representative from 3 independent experiments performed in technical duplicate. Plots followed by different letters are statistically different according to ANOVA with post-hoc Tukey's HSD.
(TIF)

**S5 Fig. Growth profile of the *A. vinelandii nifA-E356K* (EK) strain compared to the wild type (DJ) using acetate as a carbon source.** Cells were grown in MBB media supplemented with 30 mM acetate as carbon source without added ammonium (-N) or with 10 mM ammonium chloride (+N). Growth was assayed as described in S1 Fig.
(TIF)

**S6 Fig. Profile of nitrogenase expression and activity in a *nifL::*Kan$^r$ insertion strain in different carbon and nitrogen regimes.** The *nifL* disrupted strain (AZBB163) was modified to encode a *nifH::lacZ* fusion (strain 163HZ) allowing ready comparison of nitrogenase expression in (A) and nitrogenase activity in (B). Plots followed by different letters are statistically different according to ANOVA with post-hoc Tukey's HSD.
(TIF)

**S7 Fig. Levels of *nif* gene transcripts in various *P. stutzeri* strains.** (A) Diagram depicting the genotypes of the *P. stutzeri* strains analysed. Drawings are not to scale. (B) *nifH* transcripts in the strain Ps_EK compared to the wild type (Ps_WT). (C-E) levels of *nifL*, *nifA* and *nifH*

transcripts in the strain Ps_nifLA$^C$ compared to Ps_WT. (F-H) levels of *nifL*, *nifA* and *nifH* transcripts in the strain Ps_EK$^C$ compared to Ps_WT. Strains were grown under diazotrophic conditions (-N) or in the presence of excess fixed nitrogen (5 mM NH$_4$Cl, +N). In each case the data is relative to the maximum level of detected transcripts under fully derepressing conditions (Ps_WT -N) estimated from absolute quantification.
(TIF)

**S8 Fig. Growth profiles of *P. stutzeri* A1501 (wild type, Ps_WT) and the *nifA-E356K* mutant (Ps_EK$^C$) under nitrogen replete conditions.** Growth as assayed in LB media (A) or in UMS-PS medium supplemented with 30 mM glucose (B), 45 mM malate (C), 60 mM lactate (D) or in 60 mM glycerol (D). In (B-D) the nitrogen source used was 5 mM NH$_4$Cl.
(TIF)

**S9 Fig. Defining the optimal conditions for acetylene reduction and ammonia excretion in *P. stutzeri* A1501 (wild type, Ps_WT) and *nifA-E356K* mutant (Ps_EK$^C$).** (A) Acetylene reduction assay of *P. stutzeri* A1501 (wild type, Ps_WT) grown under different initial oxygen concentrations in the gas phase of batch cultures. Typical specific activities under 4% O$_2$ ranged from 18–24 nmol C$_2$H$_4$.mg protein$^{-1}$.min$^{-1}$. (B) Evaluation of the oxygen tolerance for detection of acetylene reduction upon longer incubation times for the *P.stutzeri* A1501 (Ps_WT) and *nifA-E356K* mutant (Ps_EK$^C$). Ethylene (C$_2$H$_4$) was quantified 2 hours (blue bars), 4 hours (orange bars), 6 hours (green bars) and 18 hours (yellow bars) after acetylene injection. (C) Calibration curve for ammonia quantification by the indophenol method used to quantify ammonia in the supernatant of cultures in (D), where the ammonia excretion profile at defined initial oxygen concentrations are shown. Ammonia excretion was determined after 48 hours incubation.
(TIF)

**S1 Table. Strains used in this study.**
(PDF)

**S2 Table. Plasmids used in this study.**
(PDF)

**S3 Table. Primers used in this study.**
(PDF)

# Acknowledgments

We are indebted to Brett Barney for the strain AZBB163 and to Adriano Stefanello for the plasmid pAAS1544. We are grateful to all members of the laboratory support team at the John Innes Centre for their excellent assistance.

# Author Contributions

**Conceptualization:** Marcelo Bueno Batista, Yi-Ping Wang, Ray Dixon.

**Data curation:** Marcelo Bueno Batista.

**Formal analysis:** Marcelo Bueno Batista, Paul Brett.

**Funding acquisition:** Yi-Ping Wang, Ray Dixon.

**Investigation:** Marcelo Bueno Batista.

**Methodology:** Marcelo Bueno Batista, Paul Brett, Corinne Appia-Ayme.

**Resources:** Yi-Ping Wang, Ray Dixon.

**Supervision:** Ray Dixon.

**Writing – original draft:** Marcelo Bueno Batista.

**Writing – review & editing:** Marcelo Bueno Batista, Yi-Ping Wang, Ray Dixon.

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
