## [Decision Letter · Decision Letter 0]

3 May 2021

Dear Ray,

As you will see below, the reviewers of your manuscript 'Disrupting hierarchical control of nitrogen fixation enables carbon-dependent regulation of ammonia excretion in soil diazotrophs' have expressed positive opinions (which coincide with mine). The reviewers, however, make a few suggestions to be addressed in a revised manuscript. Please modify the manuscript according to the review recommendations. Your revisions should address the specific points made by each reviewer.

[LINK]

Best regards,

Pepe

Josep Casadesús

Section Editor: Prokaryotic Genetics

PLOS Genetics

Reviewer's Responses to Questions

**Comments to the Authors:**

Reviewer #1: This is a typical Dixon paper, thoroughly thought through work with a clear message.

Overall, the work behind is technically not very complicated, but the outcome is striking: the single amino acid substitution on NifA gives rise to an ammonium excretion phenotype of the respective Azotobacter strain, and this property can be transferred to other N2-fixing Proteobacteria (with potential agricultural impact in synthetic plant -bacterial communities)

The authors provide all the necessary data to explain this phenotype, including the repression of GS activity through adenylylation. Therefore, the conclusions are backed up by the data.

There are only a few minor suggestions to improve the accessibility of the paper: it would be good to have the strain abbreviations included in the main text and not only in supplemental table:

Furthermore, the model resented in For 6 could be further improved by indicating the activity state of NifA (to easier distinguish the active state in B and F from the inactive states.

Reviewer #2: Nitrogen fixation is an energy-expensive process. For the biological reduction of nitrogen gas into ammonia, the enzyme nitrogenase utilizes 8 equivalents of ATP in each reaction cycle. The process is quite complex involved several gene products dedicated to the maturation of cofactors and enzyme components. Thus, tight regulation of this process guarantees that nitrogen fixation is only employed under conditions when no other fixed nitrogen source is available and when cells have enough nutrients to sustain this energy-demanding process. The two-component NifA-NifL transcriptional regulatory system guarantees that nitrogen fixation is activated when the right physiological and nutritional conditions are met. Notably, the NifA-NifL control guarantees that no excess of ammonia is produced, maximizing the efficiency of this system regarding the energetic toll caused by nitrogen fixation.

In previous work, the corresponding author Dixon and collaborators reported the identification of the NifA E356K variant, which was insensitive to nitrogen regulation. The E356 residue is contained within the AAA+ domain, and the screening of suppressor mutations revealed the importance of GAF and AAA+ interdomain communication for this regulatory sensing. In this study, Batista et al. further characterized this variant (NifA E356K) that bypasses the nitrogen sensing while maintaining signaling responses via 2OG.

This study builds on previous scientific contributions by Dixon and others. It provides insight into potential strategies on how to engineer microbes with expanded traits suitable for biotechnological applications in agriculture. Although the identification of this variant has been known for over 15 years, this report provides a mechanistic model for this interesting phenotype that uncouples to two regulatory features of NifA, namely nitrogen and carbon sensing.

Uncoupling nitrogen fixation from ammonia assimilation is a topic that merits investigation since it is one of the factors that limit the wide use of diazotrophs as natural fertilizers. The experimental design to characterize this variant was sounded and provided a mechanistic model for the dual regulation based on nitrogen and carbon source availability. I do not have any major comments in this submission, but I would like to provide a general suggestion for revising figure 6 and 3 so the information presented in the figure could be a little more self-explanatory. As presented, the reader needs to dig into the figure legend. Certainly, this comment does not preclude the publication of this study. Figure 3 panels shaded in green and yellow could be annotated with +N and -N. For figure 6 the suggestion is to indicate conditions for N limiting and N excess and indicate that NifA, when dissociated from the complex, is in its active form promoting transcription nitrogen fixation. Lastly, it would be good to address the literature or experimental evidence demonstrating the binding of GlnK to NifL shown in panel F. While it has been reported that GlnK directly binds to NifL, prior literature indicated that such event occurs during the formation of a ternary complex with NifA.

Reviewer #3: Overall, this is a well written and very interesting piece of work on the regulation of interaction between NifL and its cognate sigma factor NifA. It shows convincingly that a single point mutation E356K enables NifA to escape from regulation of a ternary complex with NifL and GlnK so long as 2-oxoglutarate is at sufficient concentration to bind to the GAF domain of NifA.

Reviewer specific comments

You have used the term “synthetic symbiosis” multiple times throughout the paper but fail to define what you mean by this. I assume you mean engineering control over nitrogen fixation and ammonia secretion in cereal associative bacteria? Maybe you could introduce the term in the Line 65 paragraph in the introduction.

Plant root exudates are known to be rich in dicarboxylates and other carboxylic acids, they generally do not secrete sucrose for example. You have shown in this paper that when your bacteria are grown on various dicarboxylates as sole carbon sources, NifA-E356K remains sensitive to regulation by the nitrogen status (Fig S4). This directly contradicts your claim that NifA-E356K could be useful for engineering N-insensitive diazotrophs for agricultural use. The authors should provide some acknowledgement/interpretation of this.

Specific comments

Line 61. The binding of 2-ketoglutarate by the PAS domain of NifA is such an important point that I want to see a primary reference where the measurements are reported and not a review

Page9 bottom and Figure S4. So growth on mono or di-carboxylic acids does not support such high activation of the integrated nifH-lacZ fusion. The implication is that sugars produce a higher level of 2-ketoglutarate (and this is measured), which is contrary to the simplistic expectation that solutes dependent on flux through the TCA-cycle will produce more 2-ketoglutarate. However, this may of course be a consequence of steady state levels of intermediates in the TCA-cycle being lower when growing on TCA-cycle intermediates due to a high rate of flux. I would be interested to know what genome scale models of A. vinelandii predict for intermediates of the TCA-cycle when grown on sugars or organic acids but this is clearly outside the scope of this project. However, as 2-ketoglutarate levels been measured on the different growth substrates (Fig 3) and they are higher for sugars this is consistent high TCA-cycle flux reducing the steady state concentration of 2-ketoglutarate. Perhaps a comment about flux through the TCA cycle is likely to be higher during growth on organic acids and this is likely to reduce 2-oxoglutarate levels would be helpful.

Lines 253 There is an alignment of NifA sequences in Figure 4 and this includes alpha, beta, gamma and zeta-proteobacteria but data is only given for Beta and Gamma proteobacteria, which is understandable. However, do the examples of alpha and zeta-proteobacteria have nifL? Is NifL the most common system for O2 sensing in bacteria other than gamma-proteobacteria or is it intrinsic O2 sensitivity of NifA? In other words, I would a simple but clear statement of the relative abundance of NifL in the groups of proteobacteria. This is not a trick question it just helps the reader to see how widespread the system is even if it is quite rare in some groups of bacteria.

Lines 288-295. I agree with the summary of the authors on the role of the E356K mutation. However, the rates of acetylene reduction and ammonia release, even on lactate, are extremely low for the engineered P. stuzeri. They are approximately 1% of the rates seen for A. vinelandii. From Figure S7 I see there is a very small overall escape from N-regulation in the engineered strain so essentially P. stuzeri still manages to maintain pretty good N-control of fixation in spite of the weak escape due to E356K and use of the A. vinelandii promoter. This is further discussed by the authors in the discussion.

Lines 360-362. You have shown that you do get some ammonia secretion when NifA-E356K is expressed in the glnE mutant which is unable to switch off ammonia assimilation by GS (Fig 2D-F). Thus, it is not necessary to inhibit GS to drive ammonia secretion, but it certainly enhances it.

Line 400. It might be worth mentioning here that other studies routinely assay nitrogenase activity in P. stuzeri with a starting O2 concentration of 1% (see Setten et al., 2013 & Ryu et al 2020). From the methods, your assays are done at 4% or 10% starting O2 (I am unsure which has been used). It is worth acknowledging here that you may not be observing optimal rates of nitrogen fixation.

Line 429. Is this data for comparison of phenotypes in NIL and MBB media provided? If so, could cite it here.

Line 484. Isothermal assembly? I see you mention in Line 547 that this is done with HiFi mastermix. Best to give this information when the term first appears or before-hand.

Line 555. You use the term “in brief” many times in the methods. I think this is redundant. Consider deleting.

Line 595. You state that P.stuzeri was assayed for nitrogenase activity at after adjusting the initial O2 concentration to 4% or 8%. It is not clear in your figures which was used.

**Have all data underlying the figures and results presented in the manuscript been provided?**

Reviewer #1: Yes

Reviewer #2: Yes

Reviewer #3: Yes

PLOS authors have the option to publish the peer review history of their article (what does this mean?). If published, this will include your full peer review and any attached files.

Reviewer #1: No

Reviewer #2: No

Reviewer #3: **Yes: **Phil Poole

---

## [Editor Report · Decision Letter 1]

23 May 2021

Dear Ray,

I am pleased to inform you that your manuscript entitled "Disrupting hierarchical control of nitrogen fixation enables carbon-dependent regulation of ammonia excretion in soil diazotrophs" has been editorially accepted for publication in PLOS Genetics. Congratulations!

Yours sincerely,

Pepe

Josep Casadesús

Section Editor: Prokaryotic Genetics

PLOS Genetics

Comments from the reviewers (if applicable):

**Data Deposition**

http://datadryad.org/submit?journalID=pgenetics&manu=PGENETICS-D-21-00515R1

**Press Queries**

---

## [Editor Report · Acceptance letter]

7 Jun 2021

PGENETICS-D-21-00515R1 

Disrupting hierarchical control of nitrogen fixation enables carbon-dependent regulation of ammonia excretion in soil diazotrophs 

Dear Dr Dixon, 

We are pleased to inform you that your manuscript entitled "Disrupting hierarchical control of nitrogen fixation enables carbon-dependent regulation of ammonia excretion in soil diazotrophs" has been formally accepted for publication in PLOS Genetics! Your manuscript is now with our production department and you will be notified of the publication date in due course.

With kind regards,

Katalin Szabo

PLOS Genetics

On behalf of:
